# Correlation between *Babesia* Species Affecting Dogs in Taiwan and the Local Distribution of the Vector Ticks

**DOI:** 10.3390/vetsci10030227

**Published:** 2023-03-16

**Authors:** Bi-Ling Su, Pin-Chen Liu, Jou-Chien Fang, Frans Jongejan

**Affiliations:** 1Institute of Veterinary Clinical Sciences, National Taiwan University, Taipei 106, Taiwan; r06643003@ntu.edu.tw; 2Department of Veterinary Medicine, College of Veterinary Medicine, National Chung Hsing University, Taichung 402, Taiwan; pinchenliu@dragon.nchu.edu.tw; 3Department of Veterinary Tropical Diseases, Faculty of Veterinary Science, University of Pretoria, Soutpan Road, Onderstepoort, Tshwane 0110, South Africa; frans.jongejan@up.ac.za

**Keywords:** Babesia, dogs, ticks, Rhipicephalus sanguineus, Haemaphysalis hystricis, Taiwan

## Abstract

**Simple Summary:**

Regional differences in infection rates of *Babesia* parasites affecting dogs in Taiwan were correlated with the presence of the local tick population. A positive correlation was found between pathogenic *Babesia gibsoni* infections in dogs and the tick *Haemaphysalis hystricis* in northern Taiwan. A second, less pathogenic species, *Babesia vogeli*, was more equally distributed, coinciding with the occurrence of *Rhipicephalus sanguineus,* a tick that is present throughout Taiwan. These findings provide the basis for advice for owners regarding outdoor activities with their dogs and local veterinarians with a regional diagnosis of babesiosis in Taiwan.

**Abstract:**

The objective of our study was to survey *Babesia* infection rates by PCR and tick species on stray dogs to correlate the distribution of *Babesia* with the distribution of ticks infesting dogs in Taiwan. Three hundred eighty-eight blood samples and 3037 ticks were collected from 388 roaming, and free-ranging owned dogs at residential sites in Taiwan between January 2015 and December 2017. The prevalence of *B. gibsoni* and *B. vogeli* was 15.7% (61/388) and 9.5% (37/388), respectively. Most positive *B. gibsoni* dogs were found in the northern part of the country 56/61 (91.8%), whereas a few were found in the middle 5/61 (8.2%). *Babesia vogeli* infection rates were 10%, 3.6%, and 18.2% in the northern, central, and southern regions, respectively. Five species of ticks were found: *Rhipicephalus sanguineus* (throughout Taiwan), *Rhipicephalus haemaphysaloides* (in the north), *Haemaphysalis hystricis* (in the north and middle of Taiwan), and *Amblyomma testidunarium* and *Ixodes ovatus* (both in the north). None of the dogs in the south were infected with *B gibsoni*, which correlated with the absence of *H. hystricis*, a tick recently identified as the local vector for *B gibsoni*. *Babesia vogeli* was more equally distributed, coinciding with *R. sanguineus*, a tick that is present throughout Taiwan. Anaemia was detected in 86.9% of infected dogs; among these dogs, approximately 19.7% showed severe anaemia (HCT < 20). These findings provide useful advice for owners regarding outdoor activities with their dogs and local veterinarians with a regional differential diagnosis of babesiosis in Taiwan.

## 1. Introduction

Ticks are hematophagous arthropods that parasitize vertebrates worldwide, including livestock, wildlife, and humans. Ticks play their vector roles in the transmission of a variety of pathogens. The ongoing geographic expansion of tick species, possibly driven by climatic and environmental changes, has drawn global attention [1]. However, mapping ticks and tick-borne diseases in companion animals requires further studies. For instance, babesiosis is one of the most important tick-borne infections of dogs with a worldwide distribution. *Babesia gibsoni* is the predominant tick-borne protozoan blood parasite throughout the oriental region, causing life-threatening hemolytic anaemia [2,3,4,5]. In a global distribution survey of Babesia species, Eastern Asia (Japan and South Korea) and Southeastern Asia (Singapore) showed a higher positive rate than other continents, especially *B. gibsoni* [6]. A *B. gibsoni* molecular detecting rate of 6.3% was also reported in stray dogs from Thailand [7]. Taiwan is in a subtropical region with a warm and humid climate all year round. Ticks are common ectoparasites on dogs. Therefore, tick-borne diseases in dogs have always been an important clinical topic. Canine babesiosis, including *B. gibsoni* and *B. vogeli* has been reported in dogs in Taiwan, but information on the epidemiology of the disease is limited. Several studies have focused on the prevalence of *Babesia* species in dogs in Taiwan. Chou et al. [8] collected 360 stray dogs from shelters in southern Yun-Chia-Nan and showed that the infection rate of *Babesia* species was 8.1%. Yang et al. [9] tested 265 dogs from 51 animal hospitals, and 28 dogs (10.6%) were positive for *Babesia* species, including *B. gibsoni* (26/28) and *B. vogeli* (2/28). Clinically, *B. gibsoni* cause more severe clinical symptoms and haematological abnormalities than *B. vogeli* in Taiwan.

Concerning ticks, the Asian longhorned tick, *Haemaphysalis longicornis*, is the primary vector tick in Asia for *Babesia gibsoni*. However, direct transmission from dog to dog during fighting and biting or blood transfusion has also been reported [10]. *H. longicornis* has been confirmed as an important vector of *B. gibsoni* in Japan, Korea, and China [11,12,13,14,15]. However, the tick has not been found until now in Taiwan. Earlier studies have reported that *B. gibsoni* was detected by molecular methods in different stages of the tick *Rhipicephalus sanguineus* [16]. Recently, molecular evidence for the transovarial passage of *B. gibsoni* has been reported in *Haemaphysalis hystricis* ticks from Taiwan [17]. *Rhipicephalus sanguineus* is the most common tick found on dogs in Taiwan [18], which was confirmed to play a major role in *B. vogeli* transmission [19]. Other species in relatively small numbers have also been reported on dogs in northern Taiwan, including *Rhipicephalus haemaphysaloides*, *Haemaphysalis hystricis*, *Ixodes ovatus*, *Haemaphysalis formosensis*, and *Haemaphysalis lagrangei* [18].

Canine babesiosis has attracted considerable attention due to the social impact of companion animal diseases. Several drugs and drug combinations have been described for treating acute canine babesiosis, e.g., *B. gibsoni*; however, they cannot eliminate the parasite pathogen leading to an asymptomatic carrier that may relapse and transmit the infection, even producing a drug-resistant strain [4]. Therefore, a correct diagnosis before treatment is crucial. According to previous studies [5,20], *B. gibsoni*-infected dogs had various clinical signs, such as fever, vomiting, diarrhoea, pale mucous membranes, splenomegaly, and icterus. The most common hematologic abnormalities were anaemia, thrombocytopenia, hyperglobulinemia, and hyperbilirubinemia [20]. In contrast, *B. vogeli*-naturally infected dogs generally showed asymptomatic or mild clinical signs [21,22], and a similar finding was also reported in Wang et al. [23]. The clinicopathological abnormalities were mild, except in young dogs and in adults/old dogs with predisposing factors such as splenectomy or immunocompromised condition, which can cause fever, anorexia, malaise, weight loss, regenerative anaemia, thrombocytopenia, and decreased white blood cell counts [21,22,23]. Two of three experimentally splenectomized dogs developed severe life-threatening infections [23].

Since most clinical *B. gibsoni*-infected cases occur in northern Taiwan, and veterinarians in the south rarely see infected dogs, we set out to investigate why. A greater understanding of which *Babesia* species are detected in their region may help practitioners choose appropriate testing and treatment. This study aimed to correlate the prevalence of canine *Babesia* species with the local geographical distribution of ticks infesting dogs and provide complete blood counts in *B. gibsoni* -infected stray dogs in Taiwan.

## 2. Material and Methods

### 2.1. Animal and Tick Collections

During neutering procedures, surplus blood samples and ticks were collected from 388 stray dogs at various residential sites in Taiwan between January 2015 and December 2017. The prevalence of *B. gibsoni* and *B. vogeli* was determined by PCR, whereas all ticks were identified under a stereomicroscope. All residential sites of dogs were recorded. Complete blood cell counts were determined by using an automated blood analyzer (Exigo veterinary haematological system, Boulevard Medical AB, Spånga, Sweden). A blood smear was performed after blood collection and stained by Liu’s stain (Baso, Team Medical-Tech Co., Ltd., New Taipei City, Taiwan). Platelet and differential white blood cell counts were confirmed microscopically. Direct observation of *Babesia* spp. on the blood smear was also performed microscopically. The severity of anaemia was classified and defined as mild (30% ≤ hematocrit (HCT) < 37%), moderate (20% ≤ HCT < 30%), severe (13% ≤ HCT < 20%), or very severe (HCT < 13%).

### 2.2. Multiplex-Nested PCR Amplification of the B. gibsoni 18S rRNA Gene

DNA of each dog blood sample was extracted with a genomic DNA minikit (Geneaid Biotech, Taiwan) according to the manufacturer’s instructions and stored at −20 °C until processing. All samples were tested with the PCR method published by Lin et al. [4].

The first round of PCR mixture consisted of 3 mL of 10 × Taq buffer, 0.5 mL of each primer (5′-CTACCACATCTAAGGAAGGC-3′ and 5′-TGCTTTCGCAGTAGTTCGTC-3′), 1 mL dNTPs (2.5 mM), 0.5 mL Taq DNA polymerase (GeneTeks BioScience, Inc., Taipei), and 22.5 mL of 0.1% water. First-round amplification was done as follows: three minutes of preheating at 94 °C, 35 cycles of denaturation at 94 °C for 20 s, annealing at 63 °C for 20 s and extension at 72 °C for 35 s, and the final extension at 72 °C for 5 min. The multiplex-nested PCR was performed on the first PCR product by using nested primers (5′-TGCTTTCGCAGTAGTTCGTC-3′ (*Babesia.* spp.), 5′-GTTGAATTTCTGCGTTGCCC-3′ (*B. gibsoni*), and 5′-AGTTGCCATTCGTTTGG-3′ (*B. vogeli*)). The second-round PCR amplification was performed as follows: three minutes of preheating at 94 °C, 35 cycles of denaturation at 94 °C for 20 s, annealing at 63 °C for 20 s, extension at 72 °C for 20 s, and a final extension at 72 °C for 5 min.

The first-round PCR yielded the expected products for *Babesia* spp., which was 490 bp in length. The second-round PCR yielded the expected products for *B. vogeli* and *B. gibsoni*, 249 and 268 bp in length. Positive and negative controls were added in each amplification step.

### 2.3. Phylogenetic Analysis of Babesia gibsoni from Seven Samples with Different Anaemic Severity

To analyze possible genetic fragments’ associated virulence, a total of seven samples with different anaemic severity were selected from *B. gibsoni* positive samples for sequencing, including two samples without anaemia (both from Taipei), one sample with mild anaemia (from Yilan), two samples with moderate anaemia (one sample from Taipei and the other from Nantou), and two severe anaemia samples (one from Nantou and the other from Taoyuan). The seven *B. gibsoni*-positive samples of 18s rRNA fragments were PCR-amplified by using primers 455-479F and 793-772R, described previously in Birkenheuer et al. [24]. The nucleotide sequences of target fragments were determined by using an ABI 3730XL genetic analyzer from GENOMICS (New Taipei City, Taiwan). The partial genomic sequence of seven 18srRNA of *B. gibsoni* was aligned with the other 21 18srRNA of *Babesia* spp. sequences available in the GenBank database, i.e., *B. gibsoni*: KP666168 (China), LC008284 (Bangladesh), KJ696717 (Europe), KC461261 (India), FJ554534 (Italy), AB118032 (Japan), LC012808 (Japan), AY278443 (Spain), FJ769386 (Taiwan), FJ769388 (Taiwan), AF205636 (USA), EU583386 (USA); *B. vogeli*: LC331058 (Zambia), KT438554 (Taiwan), AY072925 (Europe), HM590440 (China), AY371198 (Brazil); *Babesia canis*: KT844907 (Poland); *Babesia rossi* JN982350 (Nigeria); *Babesia conradae* AF158702 (USA). A phylogenetic analysis of partial genome sequences was performed by using the maximum-likelihood method in MEGA 10.0 software [25]. The Kimura two-parameter model analysed the genetic distance values of inter- and intraspecies variations of the *Babesia* species [26].

### 2.4. Statistical Analysis

Data obtained were analyzed by using IBM SPSS statistic version 26 (IBM Corp, Armonk, NY, USA). The results of categorical variables are expressed as percentages and presented as tables and, after that, subjected to Person’s chi-square analysis. The odds and relative risk ratios at a 95% confidence interval were used to assess the risk factors, and values of *p* < 0.05 were considered significant.

## 3. Results

A total of 3037 ticks (2742 adults, 293 nymphs and two larvae) were collected from 388 dogs at different residential locations in Taiwan. An average of 7.8 ticks was distributed over 261 dogs in North Taiwan, 83 in the middle and 44 in the south. Five species of ticks were found: *Rhipicephalus sanguineus* (2576/3037, 84.8%), *Rhipicephalus haemaphysaloides* (19/3037, 0.6%), *Haemaphysalis hystricis* (409/3037, 13.5%), *Amblyomma testudinarium* (15/3037, 0.5%), and *Ixodes ovatus* (18/3037, 0.6%). *R. sanguineus* was the predominant species on 307 of 388 dogs (79.1%), distributed throughout Taiwan. *H. hystricis* (56/388, 14.4%) was mainly distributed north of Hsinchu, east to Yilan Luodong, a small part in Puli in the central part, but not found in the south of the country. *R. haemaphysaloides* was found in the northern and central Miaoli regions. *I. ovatus* was only found in Taipei, and *A. testudinarium* on dogs in Taipei and Yilan. Sixteen dogs (4.1%) were coinfested with two species of ticks, and six dogs (1.5%) were coinfested with three different species of ticks. Seventy-seven dogs (19.8%) were infested with *H. hystricis* ticks: 96.1% (74 dogs) were from the north, whereas only 3.9% (3 dogs) were from the middle of Taiwan (Table 1). *R. sanguineus* was located on flat land at low altitudes (less than 500 m). The temperatures during collection were between 17.2 and 28.5 °C, whereas *H. hystricis* was mostly found in low-to-middle-altitude mountainous areas, with a temperature of 19.9 to 25.1 °C. *I. ovatus* was located from near sea level land to mountainous areas with an altitude of 500 to 600 m, *A. testudinarium* was found in areas from near sea level land to 100 m above sea level, with an average temperature of 22.1 °C during collection.

The prevalence of *B. gibsoni* and *B. vogeli* infection in dogs was 15.7% (61/388) and 9.5% (37/388), respectively. The regional infection rate in Taiwan’s northern, middle and southern parts were 21.6, 6 and 0%, respectively. Most positive *B. gibsoni* dogs were found in the country’s north 56/61 (91.8%), whereas a few were found in the middle 5/61 (8.2%). None of the dogs in the south was infected with *B gibsoni*. *Babesia vogeli* infection rates were 10, 3.6, and 18.2% in northern, middle, and southern regions, respectively. The infection of *B. gibsoni* in north Taiwan was significantly higher than in the middle and south (*p* < 0.001). Among the 61 dogs that were *B. gibsoni* PCR positive, only 21 (21/61, 34.4%) had *B. gibsoni* parasites in blood smears; only 2 (2/37, 5.4%) samples had *B. vogeli* parasites in blood smears among the 37 *B. vogeli* PCR positive dogs.

*B. gibsoni* infection was positively correlated with *H. hystricis*, the relative risk was 6.042 and negatively correlated with *R. sanguineus* (Table 2). Finally, the correlation between *Babesia* species infecting dogs and the distribution of vector ticks in Taiwan is shown in Figure 1.

The severities of anaemia for 388 dogs were 40.2% (156/388), 37.9% (68/388), 17.5% (68/388) and 4.4% (17/388) for nonanaemia, mild, moderate, and severe, respectively. None of the dogs had very severe anaemia (HCT < 13%). Of the 61 *Babesia gibsoni*-positive dogs, the average of hematocrit was 28.7 ± 9.1 (95% CI, 26.3–31.0), and the severities of anaemia were 13.1 (8/61), 29.5 (18/61), 37.7 (23/61), and 19.7% (12/61) for nonanaemia, mild, moderate, and severe, respectively. A total of 86.9% of infected dogs were affected by anaemia; among these dogs, approximately 19.7% showed severe anaemia (HCT < 20) (Table 3). None of the *B. gibsoni*-infected dogs were coinfected with other pathogens. Of the 37 *B. vogeli*-positive dogs, the average of hematocrit was 34.64 ± 5.5 (95% CI, 20.7–47.7). Five of them were coinfected with other pathogens (tested by PCR methods, three with *Anaplasma platys*; one with *Ehrlichia canis*; and one with Candidatus *Mycoplasma haematoparvum*). Excluding the five coinfected dogs, the average hematocrit was 35.29 ± 5.13 (95% CI, 24.4–47.7), and the severities of anaemia were 37.5% (12/32), 46.9% (15/32), and 15.6% (5/32) for nonanaemia, mild, and moderate, respectively.

Phylogenetic tree analysis is shown in Figure 2. The sequence of the seven samples showed 99–100% similarity by comparing with 13 *B. gibsoni* strains from the NCBI gene bank, 89% to *B. canis*, 89% to *B. conradae*, 86% to *B. vogeli,* and 86% to *B. rossi*.

## 4. Discussion

*Babesia* spp. were surveyed in 2012 from 360 stray dogs in the Yun-Chia-Nan area (southern Taiwan), with a positivity rate of 8.1%. However, the species of *Babesia* was not determined [8]. In 2022, Yang et al. [9] reported an increasing positive rate of 10.6%, of which 92.8% was *B. gibsoni*. In our study, the infection rate of *Babesia* spp. was generally higher, indicating that infection rates significantly increased over time. The reason may be that the sample group included stray dogs, which are more exposed to ticks, or the ongoing geographic expansion of tick species may increase the infection rate of tick-borne pathogens. In recent years, carrying domestic dogs for outdoor activities has also been increasing in Taiwan; the higher chance of natural exposure to ticks, the higher the infection rate of babesiosis.

Over 90% of positive *B. gibsoni* dogs were found in northern Taiwan, whereas only a few were found in the middle part, and none were in the south. Compared with the distribution of *H. hystricis*, it can be seen that most of the *H. hystricis* are distributed in the north, and only a small number is found in the central Nantou. None of the *H. hystricis* was found on dogs in the southern area. *H. hystrics* is a three-host tick that feeds on many mammalian species, including rodents, dogs, cattle, buffaloes, porcupines, birds, and even humans [27]. In the first survey of hard ticks infesting dogs in northern Taiwan, *R. sanguineus* (92.5%) was the dominant species, followed by *H. hystricis* (4.6%), *R. haemaphysaloides* (2.3%), *I. ovatus* (0.54%), *H. lagrangei* (0.04%), and *H. formosensis* (0.01%) [18]. In our study, we also found *Rhipicephalus sanguineus* (84.8%), *Rhipicephalus haemaphysaloides* (0.6%), *Haemaphysalis hystricis* (13.5%), and *Ixodes ovatus* (0.6%) (Table 1). The area where *H. hystricis* was collected in our study was not only in northern Taiwan, but also a small part in the middle of Taiwan, consistent with an earlier record in Puli [28]. Moreover, *Amblyomma testudinarium* was first found on dogs. The areas where *H. hystricis* are found is usually a slightly higher latitude and a lower temperature, which is more suitable for this tick species.

None of the dogs in the south of Taiwan was infected with *B. gibsoni*, which correlated with the absence of *H. hystricis*. In our study, *B. gibsoni* infection was positively associated with *H. hystricis*, with a relative risk was 6.042 (Table 2). This correlation also indirectly confirms that *H. hystricis* is the local vector of *B. gibsoni* in Taiwan and supports the earlier molecular evidence for the transovarial passage of *B. gibsoni* in *H. hystricis* ticks [17].

The infection rate of *Babesia vogeli* varied according to the region of Taiwan (Figure 1). The vector tick, *R. sanguineus* is an indoor tick (endophilic), and all stages prefer the same species of a host (monotropic) and a three-host tick. It prefers a warm and humid environment and parasitizes dogs [29]. *R. sanguineus* is the most common tick species infecting dogs in Taiwan and is distributed throughout Taiwan, and is a proven vector for *B. vogeli* in Taiwan [30,31]. *R. sanguineus* has been collected from dogs in Taipei, Taichung, Nantou, Kaohsiung, Taitung, and Hualien. The results of *R. sanguineus* distribution in this study were like those previously reported. The distribution of ticks was also consistent with the prevalence of *B. vogeli*. Interestingly, *B. gibsoni* has also been detected in *R. sanguineus* ticks by PCR [16]. Therefore, *R. sanguineus* is considered a vector of *B. gibsoni* in other world regions. However, since all *B. gibsoni*-positive specimens of adult and nymphal *R. sanguineus* ticks were collected from pet clinics in various districts of Taipei city in northern Taiwan, the tick may have fed on infected dog blood, explaining the positive PCR result.

Anaemia and thrombocytopenia were the primary haematological abnormalities in the *Babesia gibsoni*-infected dogs, consistent with previous reports [5,20]. The average hematocrit was 28.7%. About 86.9% of the dogs had anaemia, similar to Liu’s [20] survey of 60 client-owned dogs. However, the severe anaemia in this study was only approximately 20%, much less than in Liu’s study. The possible reason is that all the dogs in this study are stray dogs. If they are very anaemic (PCV < 13), they may die in the wild or have been rescued for further therapy. Our samples were collected during the neutering procedure, and the dogs showed no clinical signs (e.g., anorexia, lethargy, or inactivity) at that time. Mild anaemia was also found in *B. vogeli*-infected dogs in our study. The average hematocrit of *B. vogeli*-positive dogs was 34.64%, similar to a report about 33 naturally infected dogs in Thailand (the median of hematocrit was 35%) [22]. Even though some dogs were anaemic, there were no reported deaths at the time of surgery and within a few days after surgery.

The phylogenetic analysis of the gene fragment in this study was 100% similar to Taiwan-reported strains in the past. No new pathogenic strain was found in the study. Furthermore, there was no difference between all samples with different severities of anaemia. It is likely that the gene fragments were not associated with virulence.

Finally, this first report of *B. gibsoni* infection in stray dogs in Taiwan is limited by the relatively small number of dogs collected in the south, which may have caused some deviations in the detection rate of pathogens. Therefore, additional dogs are being monitored for this purpose.

## 5. Conclusions

More than 90% of *B. gibsoni* positive dogs were found in northern Taiwan, the hotspot of *H. hystricis*. The geographical prevalence of canine Babesia species was consistent with the distribution of the local vector ticks. The results provide helpful travel advice for dog owners and assist veterinarians with the differential diagnosis of canine babesiosis in Taiwan.

## Figures and Tables

**Figure 1 vetsci-10-00227-f001:**
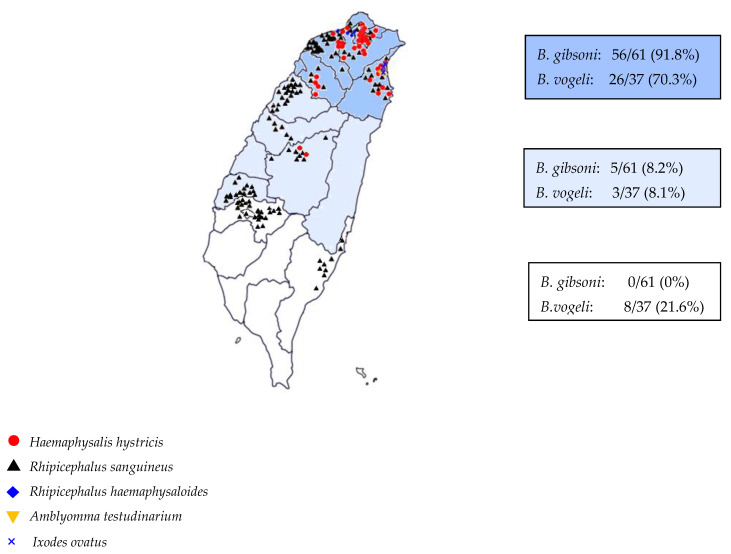
Distribution of tick species and the prevalence of *Babesia* spp. in the northern, middle and southern regions of Taiwan.

**Figure 2 vetsci-10-00227-f002:**
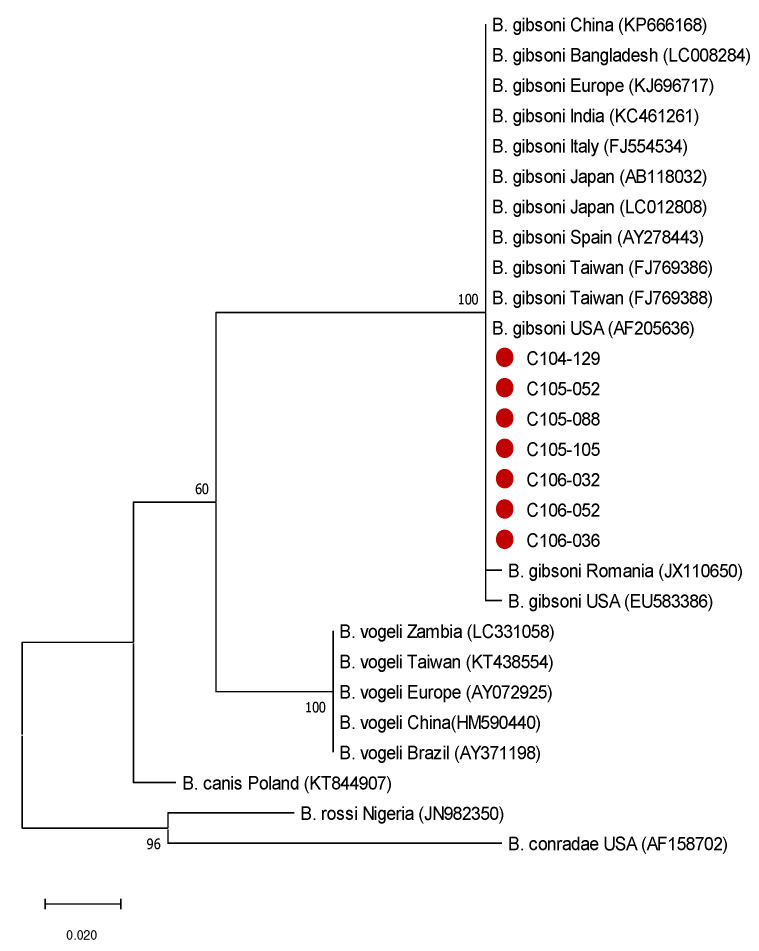
Phylogenetic tree of *B. gibsoni*. Phylogenetic relationships based on the 18S rRNA gene of *Babesia* isolates were constructed by the maximum likelihood method based on the Kimura 2-parameter model with 1000 bootstrapping replicates to determine every internal node. The red circles represent the *Babesia gibsoni* strains in this study.

**Table 1 vetsci-10-00227-t001:** The distribution of tick species collected from 388 dogs in northern, middle, and southern regions of Taiwan.

	Total	*R. sanguineus* ^a^	*H. hystricis* ^b^	*R. haemaphysaloides* ^c^	*I. ovatus* ^d^	*R. sanguineus* ^a^ + *H. hystricis* ^b^	*R. sanguineus* ^a^ + *H. hystricis* ^b^ + * A. testudinarium* ^e^	*H. hystrici* ^b^ + * A. testudinarium* ^e^	*R. sanguineus* ^a^ + *R. haemaphysaloides* ^c^	*R. sanguineus* ^a^ + *H. hystricis* ^b^ + *I. ovatus* ^d^
Northern	261	185	53	0	1	12	5	3	1	1
		(70.9%)	(20.3%)	(0%)	(0.4%)	(4.6%)	(1.9%)	(1.1%)	(0.4%)	(0.4%)
Central	83	78	3	1	0	0	0	0	1	0
		(94%)	(3.6%)	(1.2%)	(0%)	(0%)	(0%)	(0%)	(1.2%)	(0%)
Southern	44	44	0	0	0	0	0	0	0	0
		(100%)	(0%)	(0%)	(0%)	(0%)	(0%)	(0%)	(0%)	(0%)

^a^ *Rhipicephalus sanguineus*; ^b^ *Haemaphysalis hystricis*; ^c^ *Rhipicephalus haemaphysaloides*; ^d^ *I. ovatus*; ^e^ *Amblyomma testudinarium*.

**Table 2 vetsci-10-00227-t002:** (**a**) The infection rate of *Babesia* spp. in dogs infested with *H. hystricis* and other tick species. (**b**) The infection rate of *Babesia* spp. in dogs infested with *R. sanguineus* and other tick species.

(a)					
	Dogs Infested with Tick Species		*p*-Value	Relative Risk (95% CI)	Odds Ratio (95% CI)
	*H. hystricis*	Others			
*B. gibsoni*			<0.001	6.042 (3.93–9.28)	12.23 (6.2–24.15)
+	27	22			
−	29	289			
*B. vogeli*			0.326		
+	3	31			
−	53	270			
**(b)**					
	**Dogs Infested with Tick Species**		***p*-Value**	**Relative Risk** **(95% CI)**	**Odds Ratio** **(95% CI)**
	* **R. sanguineus** *	**Others**			
*B. gibsoni*			<0.001	0.456 (0.33–0.64)	0.076 (0.04–0.15)
+	21	30			
−	287	31			
*B. vogeli*			0.481		
+	31	4			
−	276	57			

**Table 3 vetsci-10-00227-t003:** Hematological variables recorded in 61 *B. gibsoni* infected dogs.

Item		Total	Nonanemia	Mild	Moderate	Severe	Reference
Number of dogs		61	8 (13.1%)	18 (29.5%)	23 (37.7%)	12 (19.7%)	
Red blood cell count	M/μL	4.62 ± 1.28	6.67 ± 0.85	5.20 ± 0.56	4.02 ± 0.58	2.62 ± 0.40	5.5–8.5
Hematocrit	%	28.7 ± 9.1	44.6 ± 7.4	33.4 ± 2.2	25.2 ± 2.9	17.14 ± 2.15	37–55
Hemoglobin	g/dL	10.38 ± 2.80	14.66 ± 2.15	11.81 ± 1.24	9.00 ± 1.15	6.13 ± 0.80	12.0–18.0
Mean corpuscular volume	fL	62.9 ± 3.94	64.3 ± 4.38	62.2 ± 2.07	62.5 ± 4.44	65.87 ± 5.84	60–72
Mean corpuscular hemoglobin	Pg	22.6 ± 1.38	21.9 ± 1.68	22.6 ± 0.94	22.6 ± 1.57	23.62 ± 1.52	19.5–25.5
Mean corpuscular hemoglobin concentration	g/dL	35.9 ± 1.51	34.2 ± 1.07	36.4 ± 1.41	36.0 ± 1.50	35.82 ± 1.34	32–38.5
Red cell distribution width	%	17.4 ± 1.67	17.7 ± 2.36	18.3 ± 5.35	17.5 ± 2.00	17.94 ± 1.39	12.0–17.5
White blood cell count	/μL	13,150 ± 5434	16,847 ± 6765	12,394 ± 5016	12,939 ± 4986.9	12,000 ± 4889	6000–17,000
Neutrophil	/μL	8967 ± 4878	11,292 ± 6525	8384 ± 4385	8869 ± 4636	8581 ± 4167	2950–11,640
Lymphocyte	/μL	2277 ± 1307	2809 ± 2705	2027 ± 893	2236 ± 993	1208 ± 790	1050–5100
Monocyte	/μL	1435 ± 955	2379 ± 1350	1539 ± 945	1191 ± 631	851 ± 562	160–1120
Eosinophil	/μL	399 ± 538	373 ± 465	424 ± 603	473 ± 582	132 ± 179	60–1230
Platelets	K/μL	169 ± 94	158 ± 68.0	185 ± 102.3	195 ± 106	155 ± 225	200–500

## Data Availability

All data are included in the current paper.

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
