# Peer review of "Correlation between Babesia Species Affecting Dogs in Taiwan and the Local Distribution of the Vector Ticks"

_vetsci, 2023, doi:10.3390/vetsci10030227_

Round 1
Reviewer 1 Report
This manuscript deals with the distribution of Babesia spp infection in dogs and its correlation with the dog ticks collected in Taiwan. However, the study is based especifaclly on Babesia gibsoni.
Canine blood samples were analized by blood smears and PCR. Ticks were collected and identified. It's missed detection of Babesia infection in those ticks collected in order to get a more accurate results. Otherwise some conclusions couldn't be supported by the results.
This manuscript should be deeply reviewed by the authors since many different mistakes and mispelling were detected.
References are not correctly quoted in the text and several mistakes have been detected such as Reference list: it lacks number (1); number (18) doesn't belong to any quote; the same for number (28).
Since the manucript title states:"...Babesia species affecting dogs in Taiwan". It lacks a deeply reviewed regarding those Babesia species, rather than B. gibsoni which is the most common. Babesia vogeli was included in the study, either in the methodology, results and discussion but there is no any information about it in the introduction section.
Are there any other small babesia in Taiwan?
Please, avoid repeated information. Line 84-87 is repeated in the MM section and in the results section (line 121-122). It should be included in just one section (it should be removed in the MM section and should be kept in the results section).
Please, include the description of blood smears in the methodology section because authors compared the results obtained by PCR and blood smears in the results section (Line 143-145)
Samples were collected from "roaming and free ranging owned dogs". How many were stray dogs and how many were owned dogs? Any statistical differences? Otherwise, it should be clarified all dogs were stray dogs.
Taking into consideration that B. gibsoni might be transovarically transmitted in ticks, it could be interesting to add information about the tick stages that were fed on positive dogs.
Table 2 should be modified in order to show results clearly.
Line 208-212. This sentence is not supported by the results, since authors do not include detection of babesia infection in ticks. Furthermore, R. sanguineus is considered a tick vector of B. gibsoni in some other regions of the world.
No clinical signs were detected in any dog. This observation is surprising since 35 out of 61 dogs B. gibsoni-infected were showing severe anemia. Were dogs clinically examined?
Reviewer 2 Report
The study detected the tick and Babesia distibution and species in dog in Taiwan. The work is very interesting and the results are important for both the research field and canine babesiosis control. However, the manuscript need improve before publish.
Major comments:
The study only described B. gibsoni and B. vogeli. There are more than 2 canine babesia species have been reported. Is there any possible that other canine babesia specie in those samples?
The same concern about tick samples as above, it might other canine babesia species in ticks in Taiwan, but the study only detected B. gibsoni and B. vogeli.
Phylogenetic analysis is needed. Highly suggest the authors sequence the 18S or ITS of B. gibsoni and B. vogeli, draw a phylogenetic tree.
M&M, results and discussion about the above should be also include.
Minor comments:
Line 29 B gibsoni correct to B. gibsoni
Line 32 correct the sentence to 86.9% of infected dogs showed anaemia
Please reformat all tables, current format is difficult to read.
Figure legend should be under the figure.
Reviewer 3 Report
The manuscript submitted for review presents as the main objective to correlate the prevalence of canine Babesia species with the local geographical distribution of ticks infesting dogs in Taiwan. A second objective was to provide complete blood counts for some B. gibsoni-infected dogs in Taiwan.
Several articles on the canine babesiosis topic have been previously reported in the literature. Nonetheless, while using available PCR-based tools for the detection and identification of two of the main canine Babesia species, pathogens of relevant clinical importance in small animal practice, the manuscript submitted provides an update on the prevalence status of B. gibsoni and B. canis vogeli, paying particular emphasis on the association of positively detected dogs with tick species identification in the infested dogs. The study found a positive correlation between B. gibsoni infection primarily with the tick vector Haemaphysalis hystricis in the northern regions of Taiwan; and B. canis vogeli with Rhipicephalus sanguineus in the middle and the northern regions of Taiwan. Overall the manuscript is well written and the most adequate citations have been included in the references section. However, minor points authors are invited to correct are the following:
Abstract
Line 20. The correct statement should be “…on stray dogs to correlate the distribution the small babesias and the infection is usually…” add “to” between “dogs" and "correlate”
Line 36. Keywords: Place all scientific names in italics.
Introduction
Line 42. Please check citation [1], in the references section, there is no reference. Either the reference is missing or the numbering is wrong, ie, should reference [2] be reference [1]? If so, all subsequent references should be re-numbered.
Line 46. Citations [2-5]. Please see the previous comment and check the citations numbering throughout the manuscript
Line 61. Place “different stage” in the plural, “different stages”
Material and methods
Line 90. Delete simbol “o” in “…of 10° x Taq…”
Line 97. Citation [3]. Check the reference. The most appropriate reference for this PCR assay is:
Birkenheuer AJ, Levy MG, Breitschwerdt EB. Development and evaluation of a seminested PCR for detection and differentiation of Babesia gibsoni (Asian genotype) and B. canis DNA in canine blood samples. J Clin Microbiol. 2003 Sep;41(9):4172-7. doi: 10.1128/JCM.41.9.4172-4177.2003.
Line 157-158. Table 1 should be presented on a separate page in a horizontal view. This is to have a better view and make it easier to read. As it is presented data in the table looks too crowded.
Line 160. Table 3. Idem. See the previous comment.
Line 160-161. Legend to figure 1 should go below the map image.
Discussion
Line 186. Correct the species name in “H. hystrics”
Line 193. Separate “H.hystricis”
Line 205. Separate “B.vogeli”
Line 209. Separate “B.vogeli”
Line 215. Cite a reference number for “Liu’s survey”
References
Please see Instructions for Authors, Manuscript Preparation, References. Make appropriate corrections for all references as indicated in the instructions for authors.
For example, Journal Articles:
1. Author 1, A.B.; Author 2, C.D. Title of the article. Abbreviated Journal Name Year, Volume, page range.
All scientific names in the titles of articles should be typed in italics. Likewise, the abbreviated journal name should be in italics. The year of publication should be bold.
References to books should cite the author(s), title, publisher, publisher location (city and country), publication year, and page.
Line 246. There is no Reference 1. Either the reference is missing or the numbering is wrong, ie, should reference [2] be reference [1]? If so, all subsequent references should be placed in the correct numerical order.
Line 284. Check reference 18. It is not a new reference. The text is part of reference 17. “18” Should be deleted. References numbering must be re-ordered and checked out according to where they were cited in the text. Place “Rhipicepahlus sanguineus” in italics.
Line 285. Check the reference number. I think it should be 17 instead of 19. Please check all subsequent references' numbering.
Line 286. Correct the term “9cks”
Line 288. Correct the term “zootiological”
Line 302. Reference 27 (and 28) should be deleted as it is repeated. It is the exact same reference as that for “17”.
Reviewer 4 Report
The manuscript is well structured, however the tables need to be corrected, and the way they are presented is confusing.
Some quote are not related to the text
Line 54 Yang et al. Add reference number
Line 97 Multiplex-nested PCR amplification of the B. gibsoni 18S rRNA gene [3]
Delete the reference number from the subtitle and place it in the description of the PCR that corresponds to this reference. Also verify if this reference number is correct in the description of the PCR methodology. Line 99 10° agregar C
Lines 100-110 â—¦C cambiar por °C
Line 106 (Babesia) cambiar a letras cursivas
The amplicons obtained from the PCR that suggest the presence of Babesia gibsoni or B. vogely were confirmed by sequencing? If so, include the % homology.
Lines171-181. This paragraph seems to me that it should be included in the introduction and not in the discussion.
References
Line 246 The reference 1 is missing
Round 2
Reviewer 1 Report
The manuscript has been highly improved. However, Babesia gibsoni could be transmitted by other routes rather than tick vector, such as bite wounds during dog fights or by transplacental transmission. It should be included in the discussion since no babesia infection was detected on ticks and samples were collected from stray dogs.
In my oppinion, results in Table 2 are important by still no clearly represented. It needs to be improved... maybe dividing it into two different tables.
Misspelling:
Line 81. In contrast instead of "contract".
Line 83: wang et al. instead of Wang et. al.
line 164-165. "In" should be written in lower case letter.
Line 183. I'd suggest to add % at the end of the value. For instance: 21.6%, 6% and 0%.. the same for Line 186, Line 205, Line 209., line 205.
Line 210. Avoid to start a sentence by using a value. For instance: Anaemia was detected in 86.9% if infected dogs.
Line 242. please correct "H. histricis".
Reviewer 2 Report
This revised manuscript is now improved over the original. The authors are thanked for addressing the concerns of the reviewer. Despite these improvements there remain English editing in need of improvement before this manuscript could be ready for publication.
